# l-Carnitine and Acetyl-l-Carnitine Induce Metabolism Alteration and Mitophagy-Related Cell Death in Colorectal Cancer Cells

**DOI:** 10.3390/nu17061010

**Published:** 2025-03-13

**Authors:** Isabella Donisi, Anna Balestrieri, Vitale Del Vecchio, Giovanna Bifulco, Maria Luisa Balestrieri, Giuseppe Campanile, Nunzia D’Onofrio

**Affiliations:** 1Department of Precision Medicine, University of Campania Luigi Vanvitelli, Via Luigi De Crecchio 7, 80138 Naples, Italy; isabella.donisi@unicampania.it (I.D.); nunzia.donofrio@unicampania.it (N.D.); 2Food Safety Department, Istituto Zooprofilattico Sperimentale del Mezzogiorno, 80055 Portici, Italy; anna.balestrieri@izsmportici.it; 3Department of Experimental Medicine, University of Campania Luigi Vanvitelli, Via Luciano Armanni 5, 80138 Naples, Italy; vitale.delvecchio@unicampania.it; 4Department of Veterinary Medicine and Animal Production, University of Naples Federico II, 80137 Naples, Italy; giovanna.bifulco@unina.it (G.B.); giuseppe.campanile@unina.it (G.C.)

**Keywords:** carnitines, colorectal cancer, SIRT4, cellular metabolism, mitophagy

## Abstract

**Background/Objectives**: Colorectal cancer (CRC) remains one of the most common and deadly malignancies worldwide, driven by metabolic reprogramming and mitochondrial dysfunction, which support tumor growth and progression. Several studies showed that nutrition is a contributing factor in the prevention and management of CRC. In this context, carnitines, amino acid derivatives abundant in food of animal origin, such as meat and milk, are crucial for mitochondrial function. Recently, l-carnitine and acetyl-l-carnitine have received particular attention due to their antioxidant, anti-inflammatory, and antitumor properties. However, to date, there is no conclusive evidence on the effects of l-carnitine and acetyl-l-carnitine in CRC or the underlying molecular mechanism. **Methods**: In this study, we investigated in HCT 116 and HT-29 CRC cells the effects of l-carnitine and acetyl-l-carnitine on mitochondrial homeostasis by XF HS Seahorse Bioanalyzer and cell death pathways by flow cytometry and western blot assays. **Results**: Data showed that l-carnitine and acetyl-l-carnitine reduced cell viability (*p* < 0.001), modulated cellular bioenergetics, and induced oxidative stress (*p* < 0.001). These phenomena promoted autophagic flux and the mitophagy process via PINK1 and Parkin modulation after 72 h of treatment. Of note, the combined treatment with l-carnitine and acetyl-l-carnitine showed a synergistic effect and enhanced the effect of single carnitines on tumor cell growth and metabolic dysfunction (*p* < 0.05). Moreover, exposure to l-carnitine and acetyl-l-carnitine promoted CRC cell apoptosis, suggesting a mechanism involving mitophagy-related cell death. These data were associated with increased SIRT4 expression levels (*p* < 0.01) and the activation of AMPK signaling (*p* < 0.01). **Conclusions**: Overall, the results, by supporting the importance of nutritional factors in CRC management, highlight l-carnitine and acetyl-l-carnitine as promising agents to target CRC metabolic vulnerabilities.

## 1. Introduction

Colorectal cancer (CRC) is one of the most prevalent malignancies worldwide and remains a leading cause of cancer-related mortality. Despite significant progress in early diagnosis and treatment, the global burden of CRC continues to rise, particularly in Western countries [1]. In fact, CRC is now the second most common cause of cancer deaths, and data suggest increasing incidence in the coming years [1]. While current therapies, such as surgery, chemotherapy, and targeted treatments, have improved survival outcomes, the prognosis remains poor, and the mortality rate trend underlines the urgent need for innovative and more effective preventive and therapeutic approaches [2].

Current studies suggest that, in addition to genetic alterations, environmental and lifestyle factors are associated with a higher risk of developing CRC [3,4,5]. Several dietary patterns and foods have been associated with CRC, given their ability to influence cellular metabolism by comprising the gut microbiome [6], impairing oxidative phosphorylation, and promoting aerobic glycolysis in a process named the *Warbug effect* [7,8]. In this context, there is growing interest in the potential roles of metabolites which the gut microbiome produces and modulates in the initiation and progression of a bioenergetic phenotype promoting uncontrolled proliferation and survival of CRC cells [5,7,8,9,10].

Carnitines are diet and endogenous metabolites derived from amino acids involved in the transport of long-chain fatty acids into the mitochondria for β oxidation and ATP production [11]. l-carnitine (Cnt) is synthesized endogenously in the liver and kidneys from lysine and methionine, but it can also be obtained from dietary sources, particularly red meats and dairy products [12]. Milk, especially milk from the Mediterranean buffalo (*Bubalus bubalis*), has been recognized as a significant contributor to dietary carnitines [13]. Compared with cow milk, Mediterranean buffalo milk contains higher levels of Cnt (2-fold) and short-chain acylcarnitines, such as acetyl-l-carnitine (C_2_Cnt, 1.4-fold), propionyl-l-carnitine (C_3_Cnt, 2.85-fold), and butyryl-l-carnitine (n-C_4_Cnt, 7-fold) [13]. Additionally, buffalo milk exhibits increased levels of betaines, including δ-valerobetaine (1.82-fold), and the Cnt precursor γ-butyrobetaine (1.5-fold), which play a crucial role in metabolic regulation and cellular protection [13].

Betaine and Cnt are precursors of trimethylamine oxide (TMAO), a compound strongly implicated in cardiovascular disease [14] and identified as a CRC dietary risk factor [15]. In contrast, recent studies have investigated the potential role of betaines, particularly milk-derived δ-valerobetaine, in CRC, highlighting its ability to counteract tumorigenic processes by modulating cellular metabolism and inducing apoptosis [16,17,18]. To date, the role of Cnt and its derivatives in CRC remain to be fully elucidated but may involve similar mechanisms.

C_2_Cnt has exhibited several pleiotropic effects in health and disease [19,20], such as notable antioxidant and anti-inflammatory properties and supporting mitochondrial integrity, which contribute to slowing down the progression of chronic diseases and aging [21,22,23].

Cnt and C_2_Cnt have also shown a potential neuroprotective role in hypoxia-ischemia, Alzheimer’s disease, and central or peripheral nervous system injury [24]. Recently, studies have implicated these metabolites in colorectal carcinogenesis, but epidemiologic evidence is conflicting thus far. Carnitine deficiency is frequently observed across a variety of metabolic and genetic disorders, including heart failure, malnutrition, diabetes, obesity, and several types of cancer [19].

Emerging evidence indicates that carnitines supplementation could also potentially influence cancer progression by modulating altered energy metabolism and reducing tumor growth by disrupting their reliance on glycolysis [20]. Cnt supplementation has demonstrated benefits in alleviating cancer-related cachexia and reducing fatigue in patients undergoing chemotherapy, enhancing overall life quality [25,26]. C_2_Cnt has shown anti-angiogenic and anti-inflammatory properties in prostate cancer cells, downregulating vascular endothelial growth factor (VEGF) and the chemokine CXCR4/CXCL12 pathways, which are critical for tumor vascularization and metastasis [27]. Additionally, C_2_Cnt has shown the ability to inhibit HepG2 and HT-29 human adenocarcinoma cell proliferation and migration, supporting its potential use as an adjunctive treatment for CRC [28]. However, to date, the role of carnitines in CRC molecular pathways remains unclear and limited.

At the molecular level, sirtuins (SIRTs), a family of class III NAD+-dependent histone deacetylases, are involved in several malignant processes, including CRC [29]. Mitochondrial SIRT4 has emerged as a crucial regulator of cellular energetic metabolism and function [30]. SIRT4 downregulation has been observed in multiple cancer types, and this deficit in CRC is associated with poor prognoses, lower survival rates, and reduced chemosensitivity [31,32,33,34,35]. SIRT4 suppressed CRC cell proliferation, migration, and invasion through inhibition of glutamine metabolism and modulating the AKT/GSK3β/CyclinD1 pathway [36,37].

In this context, considering the roles of Cnt, C_2_Cnt, and SIRT4 in cancer cellular metabolism, the present study aims to evaluate the effects of Cnt and C_2_Cnt on CRC cell lines, focusing on their potential influence on metabolic pathways in order to deepen the understanding of the CRC molecular mechanism(s) and identify new strategies for targeting the metabolic vulnerabilities of CRC.

## 2. Materials and Methods

### 2.1. Cell Growth and Treatments

Human colon epithelial CCD 841 CoN (CRL-1790) and colorectal carcinoma HCT 116 (CCL-247) and HT-29 (HTB-38) cells were purchased from ATCC (Manassas, VA, USA). CCD 841 CoN cells were maintained in Eagle’s minimum essential medium (EMEM, 30-2003, ATCC, Manassas, VA, USA), and HCT 116 and HT-29 were grown in McCoy’s 5A medium (16600-082, Gibco, Life Technologies, Carlsbad, CA, USA) supplemented with 1% penicillin-streptomycin (Pen Strep 15070063, Gibco, Life Technologies, Carlsbad, CA, USA) and 10% fetal bovine serum (FBS, 10270-106, Gibco, Life Technologies, Carlsbad, CA, USA) at 37 °C in a humidified atmosphere with 5% CO_2_.

Cnt and C_2_Cnt were purchased from Sigma-Aldrich (Milan, Italy) and dissolved in Hanks’ balanced salt solution (HBSS) with 10 mM Hepes. Cell treatments were performed by culturing cells in a complete medium with increasing concentrations of Cnt and C_2_Cnt (0–10 mM) up to 72 h. The control (Ctr) cells were treated with corresponding higher volumes of HBSS with 10 mM Hepes. When treated with *N*-acetyl-L-cysteine NAC (5 mM), the cells were pretreated for 2 h and then for 72 h in the presence of Cnt and C_2_Cnt.

### 2.2. Cell Viability Assay

Cell viability was evaluated with a Cell Counting Kit-8 (CCK-8 Donjindo Molecular Technologies, Inc., Rockville, MD, USA, 20852) following the manufacturer’s instruction. The cells were treated with Cnt and C_2_Cnt alone or in combination. Then, 10 μL of CCK-8 solution was added to each well, and the cells were incubated at 37 °C for 4 h. The absorbance was measured at 450 nm using a microplate reader (model 680, Bio-Rad, Hercules, CA, USA), and the results were reported as a percentage of Ctr (100%). All experiments were performed with *n* = 4 replicates. The combination index (CI) was formulated using the CompuSyn 1.0 software (Paramus, NJ, USA).

### 2.3. Cell Cycle Distribution

The HT-29 cells were seeded at a density of 5 × 10^4^ cells/well, while the HCT 116 cells were seeded at a density of 2.5 × 10^4^ cells/well in a 6-well plate. Prior to treatment, the cells were synchronized in the G0 or G1 phase by serum starvation, culturing them in a medium without fetal bovine serum (FBS). After synchronization, the complete medium was restored, and the cells were treated with carnitines alone or in combination for up to 72 h. After treatment, the trypsinized cells were stained with a propidium iodide (PI; Sigma Aldrich, St. Louis, MO, USA) solution (50 μg/mL PI, 25 μg/mL RNAse A, 0.1% sodium citrate, 0.1% triton in PBS) for 1 h in the dark. The intracellular DNA content was analyzed with an FACS CANTO II cytometer (BD Biosciences, San José, CA, USA), and data analysis was performed with FlowJo V10 software (FlowJo LLC, Ashland, OR, USA). At least 10,000 events were recorded for each sample.

### 2.4. Cellular Bioenergetics Analysis

Cellular bioenergetics was assessed with a Seahorse Real-Time ATP Rate Assay Kit (Agilent Technologies, Santa Clara, CA, USA). HT-29 and HCT 116 cells were seeded in a Seahorse assay microplate in growth medium. After treatment, the culture medium was replaced with complete XF DMEM (Agilent Technologies, Santa Clara, CA, USA), and the assay was performed through sequential injection of oligomycin and rotenone/antimycin A for detection the measured basal ATP production rates from mitochondrial respiration and glycolysis using an XF HS Seahorse Bioanalyzer (Agilent Technologies, Santa Clara, CA, USA).

### 2.5. Mitochondrial State

The mitochondrial state was investigated by evaluating the mitochondrial ROS production and mitophagy. Mitochondrial ROS levels were detected using MitoSOX red mitochondrial superoxide indicator (Invitrogen, Waltham, MA, USA) as already described [16]. Menadione was used as a positive control. Mitophagy was detected using a mitophagy detection kit (MD01; Dojindo Molecular Technologies, Tokyo, Japan) according to the manufacturer’s instruction. After Cnt and C_2_Cnt treatment, the HT-29 and HCT 116 cells were stained with 100 nM Mtphagy dye at 37 °C for 30 min, and the cells were subsequently incubated with 1 μM Lyso dye for an additional 30 min at 37 °C. CCCP was used as a positive control. Then, the cells were imaged on an EVOS M5000 fluorescence microscope (Thermo Scientific, Rockford, IL, USA), and fluorescence signaling was detected with an FACS CANTO II flow cytometer (BD Biosciences, San Jose, CA, USA). Data analyses were performed with FlowJo V10 software (FlowJo LLC, Ashland, OR, USA).

### 2.6. Autophagy and Cell Death Mechanism Evaluation

Autophagy was evaluated with an autophagy assay kit (ab139484, Abcam, Cambridge, UK) following the manufacturer’s indications. Briefly, the treated cells were stained with 1 µM Green Reagent for 30 min in the dark. Chloroquine was used as a positive control. The images were acquired via an EVOS M5000 fluorescence microscope (Thermo Scientific, Rockford, IL, USA). Cell death by apoptosis was evaluated with an FITC Annexin V apoptosis detection kit (BD Pharmigen, Franklin Lakes, NJ, USA, 556547) as previously reported [16]. For both essays, fluorescence was assessed using an FACS CANTO II cytometer, and the results were analyzed with FlowJo V10 software. For each sample, 20,000 events were recorded.

### 2.7. Immunoblotting

After treatment, the HT-29 and HCT 116 cells were lysed in RIPA lysis buffer (1% NP-40, 0.5% sodium deoxycholate, 0.1% SDS in PBS), and the proteins were separated via sodium dodecyl sulfate-polyacrylamide gel electrophoresis (SDS-PAGE), transferred to nitrocellulose membranes (Bio-Rad, Hercules, CA, USA), and after blocking incubated at 4 °C overnight with specific primary antibodies: anti-SIRT4 (1:1000, ab231137, Abcam, Cambridge, UK), anti-5′ AMP-activated protein kinase alpha-1 (AMPK, 1:1000, Invitrogen, Waltham, MA, USA), anti-phospho-AMPK (1:1000, Invitrogen, Waltham, MA, USA), anti-Parkin (1:2000, ab77924, Abcam, Cambridge, UK), anti-PINK1 (1:500, Y403614, Applied Biological Materials Inc., Richmond, BC, Canada), anti-microtubule-associated protein 1 light chain 3B (LC3B, 1:1000, Abcam, Cambridge, UK), anti-α-tubulin (1:3000, E-AB-20036, Elabscience Biotechnology Inc., Houston, TX, USA), anti-actin (1:5000, Abcam, Cambridge, UK), and anti-glyceraldehyde-3-phosphate dehydrogenase (GAPDH, 1:2000, ab9485, Abcam, Cambridge, UK). After incubation with secondary antibodies, the immunocomplexes were examined with the Excellent chemiluminescent substrate kit (E-IR-R301, Elabscience Biotechnology Inc., Houston, TX, USA) and visualized using the ChemiDoc Imaging System with Image Lab 6.0.1 software (Bio-Rad Laboratories, Milan, Italy). The densities of the immunoreactive bands were measured with ImageJ 1.52n software (U. S. National Institutes of Health, Bethesda, MD, USA) and expressed as arbitrary units (AUs).

### 2.8. Statistical Analysis

The results are expressed as the mean ± SD. Statistical analysis was carried out using one-way ANOVA with a Tukey post hoc test using GraphPad Prism version 9.1.2. Differences with *p* < 0.05 were considered statistically significant.

## 3. Results

### 3.1. Effect of Cnt and C_2_Cnt on CRC Viability and Cell Cycle Progression

In order to investigate the effects of Cnt and C_2_Cnt on cell growth and metabolic pathways, normal colon CCD 841 CoN and HT-29 and HCT 116 CRC cells were treated with different concentrations (0–10 mM) of Cnt and C_2_Cnt for 24 h, 48 h, and 72 h (Figure 1 and Appendix A). The results indicated that Cnt and C_2_Cnt did not show any toxic effect on non-tumor CoN CCD 841 cells, with an increase in viability observed with 10 mM Cnt (12.3 ± 6.3%, *p* < 0.05) and C_2_Cnt (17.8 ± 4.1%, *p* < 0.05) after 72 h of treatment (Appendix A). Otherwise, Cnt and C_2_Cnt selectively affected CRC cell viability in a time- and dose-dependent manner (Figure 1). In the HCT 116 cells, Cnt and C_2_Cnt showed no effect on CRC cell viability after 24 h of incubation (Figure 1A,B), with a decrease observed after 48 h of treatment with Cnt (10 mM) (13 ± 6.3%) and C_2_Cnt (5 mM) (21 ± 4.8%) (*p* < 0.05 versus Ctr) (Figure 1A,B). These effects were more consistent after 72 h of treatment with Cnt (10 mM) (30 ± 6.6%) and C_2_Cnt (5 mM) (42 ± 4.6%) (*p* < 0.001 versus Ctr) without reaching the half-maximal inhibitory concentration (IC_50_) Figure 1A,B).

In HT-29 cells, a decrease in cell viability was observed after 72 h of treatment with Cnt (10 mM) (29 ± 6.6%) and with C_2_Cnt (10 mM) (37 ± 4.8%) (*p* < 0.001 versus Ctr) (Figure 1G,H).

To explore the possible synergistic or additive effect of Cnt and C_2_Cnt, CRC cells were treated with combined carnitines (Cnt + C_2_Cnt) (Figure 1C,D,I,J). To this end, the concentration of 1 mM C_2_Cnt, inducing a reduction in CRC viability of about 30% after 72 h of treatment, was chosen for adding increasing concentrations of Cnt (up to 10 mM). The HCT 116 cells responded to the combined treatment with carnitines, reaching the IC_50_ at 1 mM C_2_Cnt combined with 5 mM Cnt (*p* < 0.001 versus Ctr). A similar trend was observed in the HT-29 cell line. The combined treatment (Cnt + C_2_Cnt) showed a synergistic effect in both the HCT 116 and HT-29 cells, as indicated by a combination index (CI) < 1 across all tested doses. In the HCT 116 cells, the strongest synergistic effect was observed at a concentration of (5 mM) Cnt + (1 mM) C_2_Cnt with a CI = 0.03602. In the HT-29 cells, the highest synergy was seen at a concentration of (1 mM) Cnt + (1 mM) C_2_Cnt with a CI = 0.13412. These findings demonstrate that the synergistic interaction between Cnt and C_2_Cnt was cell- specific and dose-dependent. Furthermore, the combined treatment was assessed using CCD 841 CoN normal colon epithelial cells. After 72 h of treatment, no reduction in cell viability was observed at the tested concentrations, confirming that the Cnt + C_2_Cnt combination selectively affected the CRC cells (Appendix A).

Based on these results, the concentrations of 10 mM Cnt and 10 mM C_2_Cnt were used for individual treatment experiments. For the combined treatment experiments, concentrations of 1 mM Cnt + 1 mM C_2_Cnt were used for the HT-29 cells, and concentrations of 5 mM Cnt + 1 mM C_2_Cnt were chosen for the HCT 116 cells.

Cell cycle distribution analysis showed the ability of Cnt and C_2_Cnt to induce cell cycle arrest in the G2 phase after 72 h of treatment (Figure 1E,F,K,L). In the HCT 116 cells, treatments with carnitines led to an increase in the G2 cell population (25.9 ± 2.2% Cnt and 30.5 ± 0.8% C_2_Cnt versus 18.8 ± 3.1% Ctr, *p* < 0.05) and a decrease in the G1 phase (40.9 ± 2.7% Cnt and 41.7 ± 3.1% C_2_Cnt versus 53.9 ± 6.6% Ctr, *p* < 0.05) (Figure 1E,F). Compared with the control cells, the combined treatment (Cnt + C_2_Cnt) resulted in the highest cell cycle G2 phase modulation (*p* < 0.001 versus Ctr), and an increase in the G2 phase was detected with respect to individual carnitines (36 ± 0.8% Cnt + C_2_Cnt versus 25.9 ± 2.2% Cnt *p* < 0.05) (36 ± 0.8% Cnt + C_2_Cnt versus 30.5 ± 0.8% C_2_Cnt, *p* < 0.05) (Figure 1E,F). In the HT-29 cells, treatment with Cnt and C_2_Cnt induced cell cycle arrest in the G2 phase (23.2 ± 0.7% Cnt and 25.3 ± 0.5% C_2_Cnt versus 12.4 ± 1.2% Ctr, *p* < 0.05) and led to a decrease in the S phase (34.9 ± 4.7% in Cnt, 38.5 ± 1.3% in C_2_Cnt versus 46.1 ± 2.1% in Ctr, *p* < 0.05). Exposure to Cnt + C_2_Cnt induced greater cell cycle arrest in the G2 phase (*p* < 0.001 versus Ctr) (33.2 ± 3.8% Cnt + C_2_Cnt versus 23.2 ± 0.7% Cnt, *p* < 0.05) (33.2 ± 3.8% Cnt + C_2_Cnt versus 25.3 ± 0.5% in C_2_Cnt, *p* < 0.05) (Figure 1K,L).

### 3.2. Cnt and C_2_Cnt Impaired the Bioenergetics of CRC Cells and Upregulated SIRT4 and AMPK

Given the crucial role of carnitines in metabolism, cellular bioenergetics was then investigated (Figure 2 and Appendix A). In the HCT 116 cells, treatment with Cnt and C_2_Cnt reduced the oxygen consumption rate (OCR) and basal glycolysis with a concomitant reduction in the total ATP production rate, consisting of the mitochondrial ATP production rate (mitoATP) and glycolysis ATP production rate (glycoATP) (*p* < 0.05) (Figure 2A–C). Similar results were observed in the HT-29 cell lines (Figure 2E–G). These effects became more pronounced following the combined treatment (*p* < 0.001 versus Ctr, *p* < 0.05 versus Cnt and C_2_Cnt).

To deepen the insight into the molecular mechanism(s) underlying the metabolic effect observed, the expression of metabolic regulators and markers of mitochondrial dynamics, namely SIRT4 and AMPK, was evaluated. The results showed an upregulation of SIRT4 in the cells treated with Cnt and C_2_Cnt (*p* < 0.01) (Figure 2D,H) and an increase in the *p*-AMPK protein levels (*p* < 0.01) (Figure 2I–N). The treatments with Cnt + C_2_Cnt showed synergistic effects, as evidenced by a greater increase in the SIRT4 and *p*-AMPK protein levels compared with individual treatments (*p* < 0.001 versus Ctr, *p* < 0.05 versus Cnt and C_2_Cnt).

### 3.3. Cnt and C_2_Cnt Promoted Mitochondrial Oxidative Stress and Mitophagy

Next, we evaluated the impact of Cnt and C_2_Cnt on the mitochondrial oxidative state in CRC cells. In the HCT 116 cells, MitoSOX staining revealed increased mitochondrial ROS levels (Cnt, *p* < 0.01 versus Ctr) (C_2_Cnt, *p* < 0.001 versus Ctr). Similarly, an increase in fluorescence intensity was induced by Cnt + C_2_Cnt (*p* < 0.001 versus Ctr, *p* < 0.01 versus Cnt, *p* < 0.05 versus C_2_Cnt) (Figure 3A–F). Furthermore, in the HT-29 cell lines, the mitochondrial ROS levels increased after stimulation with Cnt and C_2_Cnt (*p* < 0.01 versus Ctr), with the combined treatment leading to a greater increase (*p* < 0.001 versus Ctr, *p* < 0.05 versus Cnt and C_2_Cnt).

Given the linkage between mitochondrial oxidative stress and mitophagy activation [38], PINK1- and Parkin-mediated mitophagy markers were assessed (Figure 3G–P). Flow cytometric analyses revealed enhanced colocalization of mitochondria and lysosomes, with an increase in fluorescent signaling in the cells treated with Cnt (*p* < 0.01 versis Ctr in HCT 116; *p* < 0.05 versus Ctr in HT-29) and C_2_Cnt (*p* < 0.001 versus Ctr in HCT 116; *p* < 0.05 versus Crt in HT-29) and an even greater increase with Cnt + C_2_Cnt (*p* < 0.001 versus Ctr). Compared with individual carnitines, the Cnt + C_2_Cnt treatment resulted in greater red fluorescent signaling in both cell lines (*p* < 0.05 versus Cnt; *p* < 0.05 versus C_2_Cnt) (Figure 3G–I,L–N). These results were confirmed by upregulation of the expression of PINK1 and Parkin, which are key mitophagy markers (Figure 3J,K,O,P).

### 3.4. Cnt and C_2_Cnt Induced Autophagy and Apoptotic Death of CRC Cells

Exposure to Cnt and C_2_Cnt triggered autophagic flux in both the HCT 116 and HT-29 cells, as evidenced by an increase in the fluorescent signal (*p* < 0.01) accompanied by an increase in LC3B II/I expression levels (*p* < 0.05), a key autophagy marker (Figure 4A–H). Treatment with Cnt + C_2_Cnt exhibited the most substantial effect compared with the control cells (*p* < 0.001) and improved the effect of Cnt or C_2_Cnt alone (*p* < 0.05 versis Cnt and C_2_Cnt).

Treatment with Cnt and C_2_Cnt also increased apoptosis in the CRC cells (Figure 4I–L). In the HCT 116 cells, exposure to Cnt and C_2_Cnt caused a reduction in live cells (*p* < 0.01) and an increase in late (*p* < 0.001) and early apoptosis (*p* < 0.05). In the HT-29 cell line, Cnt and C_2_Cnt induced a reduction in live cells (*p* < 0.01) and an increase in necrosis (*p* < 0.05) and late and early apoptosis (*p* < 0.01). In both cell lines, the Cnt + C_2_Cnt treatment determined the greatest decrease in the live cell population (*p* < 0.001 versus Ctr) and increase in late apoptosis (*p* < 0.001 versus Ctr*)*. The combined stimulation with carnitines also enhanced the apoptotic effects of individual treatments (*p* < 0.05 versus Cnt and versus C_2_Cnt). In addition, pretreatment with the ROS scavenger NAC reduced the autophagy activation and apoptosis induced by Cnt + C_2_Cnt (Appendix A), confirming the role of oxidative stress in mediating the cell death mechanism(s) in CRC cells [16].

## 4. Discussion

The in vitro findings of this study provide the first evidence about the metabolic effects of Cnt and C_2_Cnt on CRC cells by influencing mitochondrial functions, thus promoting the mitophagy mechanism. Specifically, Cnt and C_2_Cnt treatment induced mitochondrial ROS production and impaired cellular bioenergetics by reducing mitochondrial respiration and glycolysis and promoted the activation of mitophagy and autophagic flux, leading to cell death. These effects were accompanied by the modulation of SIRT4 expression levels and the AMPK pathway, which are pivotal regulators of cellular energy homeostasis. Compared with the control cells, the in vitro combination of Cnt and C_2_Cnt displayed the greatest anti-tumor properties, supporting the synergism between carnitines and underlining the importance of considering the complex interactions among food-derived biomolecules in the setting of innovative preventive strategies for CRC. Our study highlights that the induction of apoptosis and autophagy by Cnt and C_2_Cnt is strongly linked to oxidative stress. The use of NAC, a well-known antioxidant, reduced autophagic flux and cell death, suggesting that ROS generation plays a crucial role in mediating these effects. These findings align with previous studies indicating that metabolic stress and mitochondrial dysfunction in CRC cells can promote both autophagy and apoptosis through ROS-dependent mechanisms [16].

The role of nutrition in cancer prevention and management has gained increasing attention [39]. Diet provides bioactive compounds able to modulate cancer risk and progression. CRC, characterized by its dependence on aberrant metabolic pathways, is particularly influenced by dietary components, which impact cellular metabolism and energy homeostasis [5,9]. Among milk types, buffalo milk has emerged as an important contributor in human health and in CRC due to its unique nutritional profile, which includes bioactive molecules such as carnitines (Cnt ~250 μmol/L and C_2_Cnt ~150 μmol/L) [13,16,40,41,42]. Carnitines, which are naturally present in high concentrations in animal products, play an essential role in fatty acid transport and mitochondrial energy production. These functions make them a promising candidate for dietary interventions aimed at targeting metabolic adaptations in CRC. Many studies have demonstrated that carnitines exhibit pleiotropic effects, including antioxidant and anti-inflammatory properties, contributing to their potential therapeutic applications across various pathological conditions. In metabolic disorders, Cnt supplementation has shown significant benefits in obesity induced by a high-fat diet (HFD), improving lipid profiles by reducing serum triglyceride (TG) and glutamic-oxaloacetic transaminase (GOT) levels [43,44]. In the management of diabetes, Cnt administration has been reported to enhance glycogen storage and glucose oxidation in patients with type 2 diabetes, where plasma Cnt levels are often reduced [19,45]. In addition, Cnt supplementation also reduced serum lipids and plasma glucose in patients with type 2 diabetes, suggesting its potential role in ameliorating metabolic control in diabetic patients [46]. C_3_Cnt and C_2_Cnt possess cardioprotective properties and have been recommended for conditions such as reperfusion injury, coronary infarction, arrhythmias, and myocardial damage [19]. Meta-analyses indicated that carnitine supplementation reduces arrhythmias, ventricular dysfunction, and angina-related pain, ultimately lowering the risk of myocardial infarction and mortality [47]. Moreover, carnitines have shown neuroprotective properties in neurodegenerative diseases, such as Alzheimer’s and Parkinson’s disease, by reducing neuroinflammation and mitigating mitochondrial dysfunction [48].

Several studies previously highlighted the potential of carnitine derivates in cancer biology [20]. Reduced total carnitines serum levels were found in women with endometrial cancer, and these levels decreased progressively with advancing tumor stages [49]. Carnitine levels were also found to be decreased in pediatric cancer patients and in colorectal cancer tissue compared with normal tissue [50,51]. The role of carnitines has been widely explored in cancer cachexia, where Cnt and C_2_Cnt supplementation has shown beneficial effects through reducing the serum concentrations of the cytokines inteleukin (IL)-6 and tumor necrosis factor (TNF)-α and the downstream molecules of the NF-κB and Cox-2 inflammatory pathways [52,53,54,55]. In addition, the administration of 100 mg/kg Cnt showed inhibitory effects on the initiation of hepatocarcinogenesis in mice [56].

In agreement with previous in vitro and in vivo studies indicating the ability of carnitines to inhibit cancer cell proliferation and induce apoptosis [27,28,57,58], our results show the capacity of Cnt and C_2_Cnt to reduce cell viability and induce apoptosis in CRC cells. Cnt treatment modulated mitochondria-dependent apoptosis pathways in hepatoma cells by inducing the upregulation of caspase-9 and caspase-3 [58]. In prostate cancer, C_2_Cnt induces apoptosis and prevents production of the pro-inflammatory cytokines TNF-α and interferon (IFN)-γ. In addition, C_2_Cnt reduced the release of the pro-angiogenic factor VEGF and the production of chemokines involved in adhesion, migration, and invasion [27]. In CRC cells, other studies have shown that C_2_Cnt is able to induce a reduction in cell viability, affect the wound healing capacity, and reduce matrix metalloproteinase-9 (MMP-9) and VEGF expression levels, suggesting the ability of C_2_Cnt to counteract the adhesion, migration, and invasion of HT-29 cells [28]. Moreover, palmitoylcarnitine, together with Cnt, potently induces apoptosis in HT-29 cells [57], and carnitine cycle dysfunction has been shown to influence carcinogenesis and tumor progression by regulating metabolic reprogramming [59]. In contrast, one study suggested that Cnt supplementation (50 µM) in HCT 116 cells increases butyrate oxidation, promoting cancer cell survival by reducing its epigenetic impact and potentially limiting its role as an HDAC inhibitor while reducing histone acetylation [60].

Our study added new insights on the role of Cnt and C_2_Cnt in modulating CRC cell metabolism, showing that their single supplementation as well as combined Cnt + C_2_Cnt treatment disrupt cellular bioenergetics by decreasing mitochondrial respiration and glycolysis, a determinant feature of tumor survival and progression. Of note, combined treatment with carnitines highlighted the potential synergy between these derivatives in targeting CRC cells. The observed synergy could result from their complementary actions, where Cnt is a key derivative involved in cellular metabolism, while C_2_Cnt influences epigenetic pathways due to its acetyl group donor properties [61]. Although both compounds independently modulated metabolic pathways and induced cell death, their combined use amplified these effects, making combined treatment an attractive strategy for the management of CRC.

All of these events were accompanied by upregulation of the SIRT4 levels and *p*-AMPK activation, in accordance with other works which demonstrated the oncosuppressor role of SIRT4 in many cancers, including CRC. In CRC, the levels of SIRT4 are decreased compared with normal colon tissue, and its lower levels are associated with worse prognoses and lower survival rates in patients, while SIRT4 overexpression inhibits CRC cell proliferation in vitro and in vivo, inhibiting glutamine metabolism and glycolysis [32,34]. A recent study by Li et al. demonstrated that SIRT4 suppressed the tumor growth of pancreatic ductal adenocarcinoma by inducing autophagy and activating *p*-p53 protein and *p*-AMPKα [62]. In our study, carnitine treatment was able to induce mitophagy and autophagy, as evidenced by increased expression levels of the key markers PINK1 and Parkin and the LC3BII/I ratio. AMPK is generally phosphorylated to restore energy metabolism when ATP synthesis is impaired. In CRC, AMPK deficiency aggravates tumorigenesis in vivo, while its activation by SIRT3 reduces cell viability and induces ferroptosis in HCT 116 cells via SIRT3 [63,64]. Undoubtedly, this study lacks detailed investigation into the downstream signaling pathways of SIRT4 and AMPK, including mTOR. Recent evidence suggests that SIRT4 may act as a metabolic checkpoint by repressing glutamine metabolism and interacting with mTORC1, thereby influencing cancer cell growth and stress adaptation [65]. Similarly, AMPK activation is known to suppress mTOR signaling while promoting pathways such as autophagy and apoptosis [66]. The interplay between these factors remains a critical area for further investigation, as understanding these mechanisms may provide novel insights into potential therapeutic strategies targeting the SIRT4-AMPK-mTOR signaling axis.

In our in vitro experimental model, the concentrations of 10 mM Cnt and 10 mM C_2_Cnt determined an alteration of the metabolic pathway, which could explain the anti-tumor effects, with differences between the two cell lines, likely due to the p53 status. The HCT 116 cells expressed wild-type p53, which is known to act as a tumor suppressor by promoting apoptosis, regulating mitochondrial metabolism, and enhancing autophagy in response to cellular stress [67,68]. Conversely, the HT-29 cells harbored a mutant form of p53 often associated with a loss of tumor suppressive functions and a shift toward metabolic reprogramming, affecting cell survival and energy homeostasis [67,68].

It is important to point out that at this concentration, although above the normal plasma levels in humans, Cnt and C_2_Cnt are not toxic to non-tumor cells. In humans, Cnt and C_2_Cnt accumulate in the kidneys, liver, and brain, being absent from skeletal muscle and the heart and reaching plasma concentrations in the range of 25–50 μM for Cnt and 3–6 μM for C_2_Cnt [11]. Cnt and C_2_Cnt are mostly obtained from food of animal origins and vegetables, with the latter being rich in *N*(ε)-trimethyllysine, the precursor of carnitine biosynthesis [69]. Undoubtedly, there is a need to consider the appropriate amount of carnitine supplementation while also considering the possible biological effects which can derive from interaction with other biomolecules. Although evidence regarding the safety of dietary interventions or supplements with Cnt and C_2_Cnt and in CRC patients is lacking, it is plausible to speculate that long-term supplementation associated with oral administration of Cnt and C_2_Cnt could affect the plasma carnitine content, thus shifting the CRC metabolic pathways. However, it is advisable to promote further studies in animal models and clinical trials to evaluate whether dietary interventions or supplements with Cnt and C_2_Cnt can selectively modulate tumor metabolism without triggering adverse effects related to microbiota-derived metabolites [15]. Moreover, while the upregulation of SIRT4 was observed, further investigations using genetic tools, such as SIRT4 silencing or overexpression, are needed to confirm its role. These results support the rationale for exploring the real efficacy of dietary carnitine supplementation as potential metabolic modulators in CRC and as an adjuvant strategy in the prevention and treatment of CRC.

## 5. Conclusions

In conclusion, our results reinforce the growing recognition of nutrition as a key factor in CRC prevention and management by showing new evidence on the potential antitumor effects of Cnt and C_2_Cnt, compounds which are highly present in milk and dairy products. These results also support Cnt + C_2_Cnt as a synergic combination for innovative strategies in CRC prevention. By targeting the metabolic vulnerabilities of CRC cells through the modulation of mitochondrial function and SIRT4 activity, Cnt + C_2_Cnt could offer promising aid to existing approaches in precision oncology.

## Figures and Tables

**Figure 1 nutrients-17-01010-f001:**
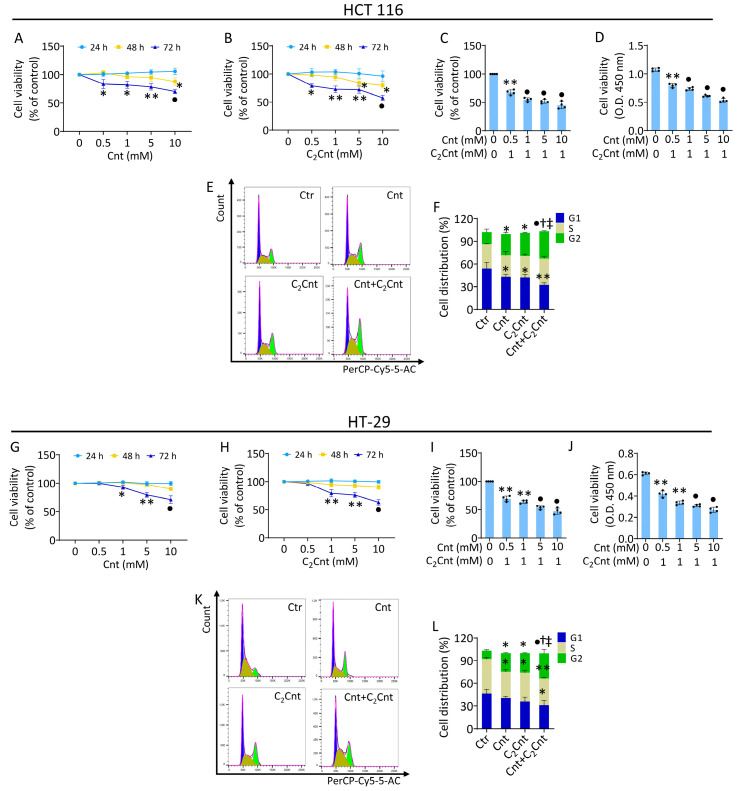
Effects of Cnt and C_2_Cnt on CRC cell viability. Cell viability assessed in HCT 116 and HT-29 cells stimulated with different concentrations of (**A**–**G**) Cnt and (**B**–**H**) C_2_Cnt (0–10 mM) for 24, 48, and 72 h. Cell viability measured after 72 h of treatment with 1 mM C_2_Cnt and increasing concentrations of Cnt (0–10 mM) in HCT116 (**C**,**D**) and HT-29 cells (**I**,**J**). Results of cell viability were reported as % of control and Optical Density (O.D.). Representative cell cycle analysis with FACS in (**E**,**F**) HCT 116 and (**K**,**L**) HT-29 cells exposed to carnitines. Control cells were grown in medium containing the same volume of HBSS with 10 mM Hepes. The data are expressed as the mean ± SD of *n* = 4 independent experiments. * *p* < 0.05, indicating significant differences between 0 μg/mL (or control) and sample treatments (Cnt/C_2_Cnt). ** *p* < 0.01, indicating significant differences between 0 μg/mL (or control) and sample treatments (Cnt/C_2_Cnt). • *p* < 0.001, indicating significant differences between 0 μg/mL (or control) and sample treatments (Cnt/C_2_Cnt). ^†^ *p* < 0.05, indicating significant differences between Cnt and combined treatments (Cnt + C_2_Cnt). ^‡^ *p* < 0.05, indicating significant differences between C_2_Cnt and combined treatments (Cnt + C_2_Cnt).

**Figure 2 nutrients-17-01010-f002:**
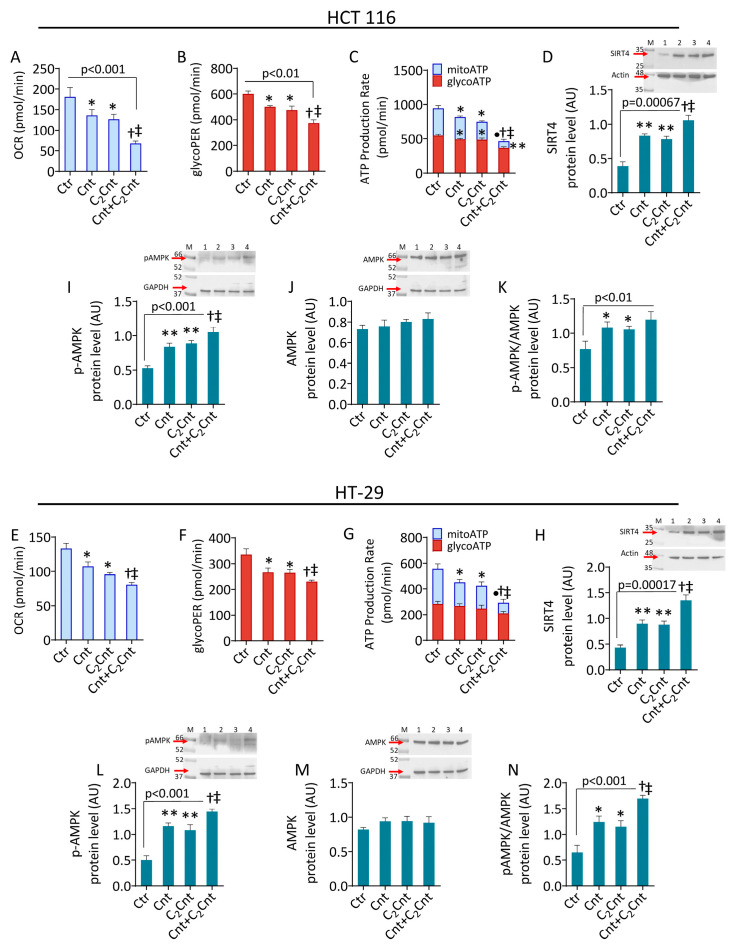
Cnt and C_2_Cnt impaired CRC cell metabolism and modulated SIRT4 levels. Seahorse analysis of oxygen consumption rate, glycolytic proton efflux rate, and ATP production rate measured in (**A**–**C**) HCT 116 and (**E**–**G**) HT-29 cells stimulated with Cnt and C_2_Cnt alone or in combination for 72 h. Immunoblotting analysis with cropped blots of (**D**,**H**) SIRT4, (**I**,**L**) *p*-AMPK, and (**J**,**M**) AMPK protein levels and (**K**,**N**) *p*-AMPK/AMPK ratio in CRC cells treated with carnitines for 48 h. M = molecular weight markers; lane 1 = Ctr; lane 2 = Cnt; lane 3 = C_2_Cnt; lane 4 = Cnt + C_2_Cnt. Data are expressed as mean ± SD of *n* = 3 experiments. * *p* < 0.05, indicating significant differences between control (Ctr) and sample treatments (Cnt/C_2_Cnt). ** *p* < 0.01, indicating significant differences between control (Ctr) and sample treatments (Cnt/C_2_Cnt). • *p* < 0.001, indicating significant differences between control (Ctr) and sample treatments (Cnt/C_2_Cnt). ^†^ *p* < 0.05, indicating significant differences between Cnt and combined treatments (Cnt + C_2_Cnt). ^‡^ *p* < 0.05 indicating significant differences between C_2_Cnt and combined treatments (Cnt + C_2_Cnt).

**Figure 3 nutrients-17-01010-f003:**
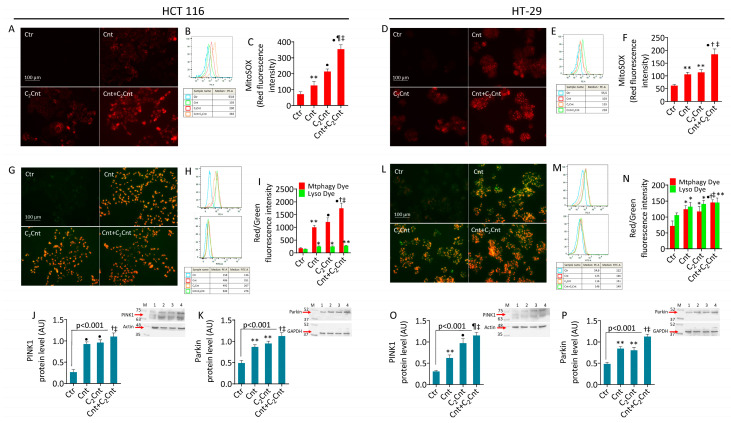
Cnt and C_2_Cnt induced mitochondrial damage. Representative fluorescent images and FACS analysis of (**A**–**F**) mitochondrial ROS and (**G**–**N**) mitophagy and immunoblotting analysis with cropped blots of (**J**,**O**) PINK1 and (**K**,**P**) Parkin protein levels in HCT 116 and HT-29 cells stimulated with carnitines alone or in combination for 72 h. Results are expressed as median fluorescence intensity (MFI). Data are expressed as mean ± SD of *n* = 3 experiments. Scale bars = 100 μm. M = molecular weight markers; lane 1 = Ctr; lane 2 = Cnt; lane 3 = C_2_Cnt; lane 4 = Cnt + C_2_Cnt. * *p* < 0.05, indicating significant differences between control (Ctr) and sample treatments (Cnt/C_2_Cnt). ** *p* < 0.01, indicating significant differences between control (Ctr) and sample treatments (Cnt/C_2_Cnt). • *p* < 0.001, indicating significant differences between control (Ctr) and sample treatments (Cnt/C_2_Cnt). ^†^ *p* < 0.05, indicating significant differences between Cnt and combined treatments (Cnt + C_2_Cnt). ^¶^ *p* < 0.01, indicating significant differences between Cnt and combined treatments (Cnt + C_2_Cnt). ^‡^ *p* < 0.05 indicating significant differences between C_2_Cnt and combined treatments (Cnt + C_2_Cnt).

**Figure 4 nutrients-17-01010-f004:**
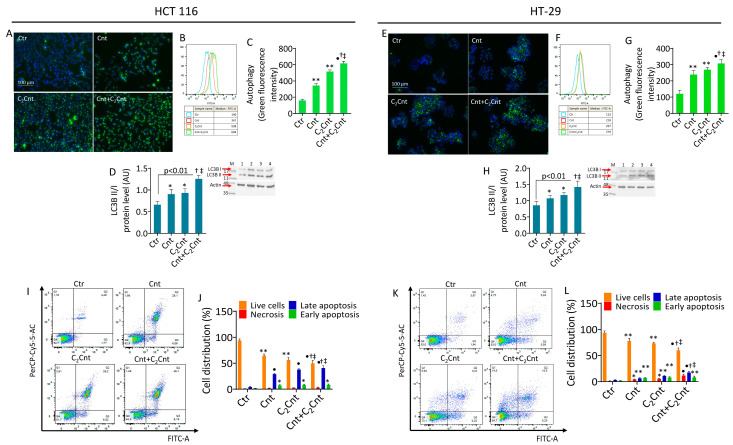
Cnt and C_2_Cnt promoted autophagic flux and cell death. Representative fluorescent images and cytofluorimetric detection of autophagy and immunoblotting analysis with cropped blots of LC3B II/I ratio in (**A**–**D**) HCT 116 and (**E**–**H**) HT-29 cell lines treated with Cnt and C_2_Cnt alone or in combination for 72 h. (**I**–**L**) Dot plots and analyses of annexin V-FITC and PI staining. Data are expressed as mean ± SD of *n* = 3 experiments. Q1 = necrotic cells; Q2 = late apoptotic cells; Q3 = early apoptotic cells; Q4 = viable cells. M = molecular weight markers; lane 1 = Ctr; lane 2 = Cnt; lane 3 = C_2_Cnt; lane 4 = Cnt + C_2_Cnt. Scale bars = 100 μm. * *p* < 0.05, indicating significant differences between control (Ctr) and sample treatments (Cnt/C_2_Cnt). ** *p* < 0.01, indicating significant differences between control (Ctr) and sample treatments (Cnt/C_2_Cnt). • *p* < 0.001, indicating significant differences between control (Ctr) and sample treatments (Cnt/C_2_Cnt). ^†^ *p* < 0.05, indicating significant differences between Cnt and combined treatments (Cnt + C_2_Cnt). ^‡^ *p* < 0.05 indicating significant differences between C_2_Cnt and combined treatments (Cnt + C_2_Cnt).

## Data Availability

The original contributions presented in the study are included in the article and Appendix A, further inquiries can be directed to the corresponding author.

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
