# Peer review of "l-Carnitine and Acetyl-l-Carnitine Induce Metabolism Alteration and Mitophagy-Related Cell Death in Colorectal Cancer Cells"

_nutrients, 2025, doi:10.3390/nu17061010_

Round 1
Reviewer 1 Report
Comments and Suggestions for Authors
This study investigates the anticancer effects of L-Carnitine, Acetyl-L-carnitine, and their combination against human colorectal cancer, while also elucidating the underlying mechanisms. The authors have conducted a well-designed and thoroughly executed study, and the manuscript is generally well-written. However, there are a few minor typographical and formatting issues that need to be addressed. After these corrections, the manuscript has the merit for publication in Nutrients.
Comments:
- Line 139: Please specify the initial cell seeding density for the 6-well plate to ensure reproducibility.
- Line 140: Indicate whether the cells were synchronized prior to sample treatment, as this impacts the interpretation of results.
- Figures: The spacing between panels is insufficient, which may hinder readability. I suggest adjusting the layout to provide clearer separation between each panel.
- Line 265: The phrase "p < 0.05 vs 0 μg/mL or Ctr" is unclear. Consider revising it to: "p < 0.05, indicating significant differences between 0 μg/mL (or Control) and sample treatments (ctn/c2ctn)." Ensure this clarification is consistently applied across all figure legends.
- Formatting: According to MDPI guidelines:
- Figures in the main text should be referred to as Figures rather than Fig.
- Titles and subtitles should follow title case, with each word starting with a capital letter.
Addressing these points will enhance the clarity and professionalism of the manuscript.
Author Response
nutrients-3511346
Response to reviewers’ comments
We thank the referees for the helpful comments on the manuscript. We have addressed the issues according to the reviewer’s comments/suggestions. We believe that these modifications have improved the paper. Changes are highlighted in the revised version of the manuscript.
Reviewer #1:
This study investigates the anticancer effects of L-Carnitine, Acetyl-L-carnitine, and their combination against human colorectal cancer, while also elucidating the underlying mechanisms. The authors have conducted a well-designed and thoroughly executed study, and the manuscript is generally well-written. However, there are a few minor typographical and formatting issues that need to be addressed. After these corrections, the manuscript has the merit for publication in Nutrients.
Reply: We thank the Reviewer for the helpful comments on the manuscript. The revision has been performed according to Reviewer’ requests. The helpful comments have undoubtedly contributed to strengthen the manuscript (revision marked in yellow).
Comments:
- Line 139: Please specify the initial cell seeding density for the 6-well plate to ensure reproducibility.
Reply: Authors thank the Reviewer for the suggestion. As requested, we have specified the initial cell seeding density for the 6-well plate to ensure reproducibility: " HT-29 cells were seeded at a density of 5 × 10⁴ cells/well, while HCT 116 cells were seeded at a density of 2.5 × 10⁴ cells/well in a 6-well plate"
Please, see lines 141-142
- Line 140: Indicate whether the cells were synchronized prior to sample treatment, as this impacts the interpretation of results.
Reply: Authors thank the Reviewer for the comment. “Prior to treatment, cells were synchronized in the G0/G1 phase by serum starvation, culturing them in medium without fetal bovine serum (FBS). After synchronization, complete medium was restored, and cell were treated with carnitines alone or in combination up to 72h”.
Please, see lines 142-145
- Figures: The spacing between panels is insufficient, which may hinder readability. I suggest adjusting the layout to provide clearer separation between each panel.
Reply: Authors appreciate the Reviewer’s suggestion. The layout and spacing between panels have been adjusted to improve readability in revised Manuscript.
Please, see Figure 1, 2, 3 and 4 and Supplementary Figure S1, S2 and S3.
- Line 265: The phrase "p < 0.05 vs 0 μg/mL or Ctr" is unclear. Consider revising it to: "p < 0.05, indicating significant differences between 0 μg/mL (or Control) and sample treatments (ctn/c2ctn)." Ensure this clarification is consistently applied across all figure legends.
Reply: Authors thank the reviewer for the suggestion. According, the phrasing has been revised to: “*p < 0.05, indicating significant differences between 0 μg/mL (or Control) and sample treatments (Cnt/C2Cnt); ** p < 0.01, indicating significant differences between 0 μg/mL (or Control) and sample treatments (Cnt/C2Cnt); • p < 0.001, indicating significant differences between 0 μg/mL (or Control) and sample treatments (Cnt/C2Cnt); † p < 0.05, indicating significant differences between Cnt and combined treatments (Cnt+C2Cnt); ‡ p < 0.05 indicating significant differences between C2Cnt and combined treatments (Cnt+C2Cnt).”
Please, see lines 271-276
“*p < 0.05, indicating significant differences between Control (Ctr) and sample treatments (Cnt/C2Cnt); ** p < 0.01, indicating significant differences between Control (Ctr) and sample treatments (Cnt/C2Cnt); • p < 0.001, indicating significant differences between Control (Ctr) and sample treatments (Cnt/C2Cnt); † p < 0.05, indicating significant differences between Cnt and combined treatments (Cnt+C2Cnt); ‡ p < 0.05 indicating significant differences between C2Cnt and combined treatments (Cnt+C2Cnt).”
Please, see lines 301-306
“*p < 0.05, indicating significant differences between Control (Ctr) and sample treatments (Cnt/C2Cnt); ** p < 0.01, indicating significant differences between Control (Ctr) and sample treatments (Cnt/C2Cnt); • p < 0.001, indicating significant differences between Control (Ctr) and sample treatments (Cnt/C2Cnt); † p < 0.05, indicating significant differences between Cnt and combined treatments (Cnt+C2Cnt); ¶ p < 0.01, indicating significant differences between Cnt and combined treatments (Cnt+C2Cnt); ‡ p < 0.05 indicating significant differences between C2Cnt and combined treatments (Cnt+C2Cnt).”
Please, see lines 333-339
“*p < 0.05, indicating significant differences between Control (Ctr) and sample treatments (Cnt/C2Cnt); ** p < 0.01, indicating significant differences between Control (Ctr) and sample treatments (Cnt/C2Cnt); • p < 0.001, indicating significant differences between Control (Ctr) and sample treatments (Cnt/C2Cnt); † p < 0.05, indicating significant differences between Cnt and combined treatments (Cnt+C2Cnt); ‡ p < 0.05 indicating significant differences between C2Cnt and combined treatments (Cnt+C2Cnt).”
Please, see lines 366-371
“*p < 0.05, indicating significant differences between 0 μg/mL (or Control) and sample treatments (Cnt/C2Cnt).”
Please, see figure legend of Figure S1.
- Formatting: According to MDPI guidelines:
Figures in the main text should be referred to as Figures rather than Fig.
Titles and subtitles should follow title case, with each word starting with a capital letter.
Reply: We appreciate the Reviewer’s comment. In the revised Manuscript, 'Fig.' has been replaced with 'Figure' throughout the text, and titles and subtitles now follow title case.
Addressing these points will enhance the clarity and professionalism of the manuscript.
Reply: All suggestions have been addressed to the fullest.
Reviewer 2 Report
Comments and Suggestions for Authors
This study investigates the impact of L-carnitine (Cnt) and acetyl-L-carnitine (C2Cnt) on colorectal cancer (CRC) cells, focusing on their potential as metabolic modulators and therapeutic agents. The research highlights the role of these compounds in mitochondrial function, oxidative stress, and cell death through mitophagy. The authors demonstrate that Cnt and C2Cnt inhibit CRC cell viability, induce oxidative stress, and promote autophagy-mediated apoptosis. Additionally, the study shows that their combined treatment enhances synergistic effect, enhancing tumor cell death. The study also links these effects to the upregulation of SIRT4 and AMPK signaling, suggesting that these pathways play a crucial role in the metabolic alterations observed in CRC cells. The findings suggest that dietary interventions with these compounds may serve as potential adjunctive strategies in CRC management. However, there are still some issues that need to be addressed.
- While the study highlights the involvement of SIRT4 and AMPK in metabolic regulation, it doesn’t thoroughly explore downstream signaling pathways affected by these proteins.
- The paper doesn’t investigate the toxicity of combined carnitines in normal cells.
- Were there any differences in how HCT 116 (p53 wild-type) and HT-29 (p53 mutant) CRC cells responded to treatment, given that p53 influences mitochondrial metabolism and autophagy?
- The study mentions that Cnt and C2Cnt increase oxidative stress, leading to CRC cell death. However, was a ROS scavenger, such as N-acetylcysteine (NAC), used to confirm that oxidative stress is a direct cause of cell death?
- Check the data in the control group of Figure 1C and 1H. Each point means data in the control group. The four data of the control group can’t be the same.
- Compared to Figure 3G with 3L, the background of the confocal images is different. It’s better to keep the same conditions for every confocal image.
- It is recommended to enlarge Figure 4J and 4L as it is difficult to read the details.
Author Response
nutrients-3511346
Response to reviewers’ comments
We thank the referees for the helpful comments on the manuscript. We have addressed the issues according to the reviewer’s comments/suggestions. We believe that these modifications have improved the paper. Changes are highlighted in the revised version of the manuscript.
Reviewer #2:
This study investigates the impact of L-carnitine (Cnt) and acetyl-L-carnitine (C2Cnt) on colorectal cancer (CRC) cells, focusing on their potential as metabolic modulators and therapeutic agents. The research highlights the role of these compounds in mitochondrial function, oxidative stress, and cell death through mitophagy. The authors demonstrate that Cnt and C2Cnt inhibit CRC cell viability, induce oxidative stress, and promote autophagy-mediated apoptosis. Additionally, the study shows that their combined treatment enhances synergistic effect, enhancing tumor cell death. The study also links these effects to the upregulation of SIRT4 and AMPK signaling, suggesting that these pathways play a crucial role in the metabolic alterations observed in CRC cells. The findings suggest that dietary interventions with these compounds may serve as potential adjunctive strategies in CRC management. However, there are still some issues that need to be addressed.
Reply: Authors deeply thank the Reviewer for the careful revision and critical understanding of the manuscript. The revision has been performed according to Reviewer’ requests. The critical suggestions have undoubtedly contributed to strengthening the manuscript (revision marked in blue).
- While the study highlights the involvement of SIRT4 and AMPK in metabolic regulation, it doesn’t thoroughly explore downstream signaling pathways affected by these proteins.
Reply: Authors thank the Reviewer for the insightful suggestion. The limitation of the study that doesn’t thoroughly explore downstream signaling pathways affected by SIRT4 and AMPK has been discussed in the revised Manuscript. “Undoubtedly, this study lacks of detailed investigation into the downstream signaling pathways of SIRT4 and AMPK, including mTOR. Recent evidence suggests that SIRT4 may act as a metabolic checkpoint by repressing glutamine metabolism and interacting with mTORC1, thereby influencing cancer cell growth and stress adaptation [65]. Similarly, AMPK activation is known to suppress mTOR signaling while promoting pathways such as autophagy and apoptosis [66]. The interplay between these factors remains a critical area for further investigation, as understanding these mechanisms may provide novel insights into potential therapeutic strategies targeting the SIRT4-AMPK-mTOR signaling axis.”
Please see Discussion section, lines 472-480 and new reference 65 and 66.
- The paper doesn’t investigate the toxicity of combined carnitines in normal cells.
Reply: According to Reviewer comment, data showing the effects of combined carnitine treatment in normal cells have been reported in the revised Manuscript. “Furthermore, the combined treatment was assessed on normal colon epithelial cells CCD 841 CoN. After 72 h of treatment, no reduction in cell viability was observed at the tested concentrations, confirming that the Cnt+C2Cnt combination selectively affects CRC cells (Figure S1C,D).”
Please see Results section, lines 242-245 and new Supplementary Figure S1, Panel C and Panel D.
- Were there any differences in how HCT 116 (p53 wild-type) and HT-29 (p53 mutant) CRC cells responded to treatment, given that p53 influences mitochondrial metabolism and autophagy?
Reply: Authors thank the Reviewer for the comment. According, the different responses to treatment between HCT 116 (p53 wild-type) and HT-29 (p53 mutant) CRC cells have been provided in the revised Manuscript. “In our in vitro experimental model, the concentrations of 10 mM Cnt and 10 mM C2Cnt determined an alteration of the metabolic pathway, which could explain the anti-tumour effects, with differences between the two cell lines, likely due to the p53 status. HCT 116 cells express wild-type p53, which is known to act as a tumor suppressor by promoting apoptosis, regulating mitochondrial metabolism, and enhancing autophagy in response to cellular stress [67, 68]. Conversely, HT-29 cells harbor a mutant form of p53, often associated with a loss of tumor suppressive functions and a shift towards metabolic reprogramming affecting cell survival and energy homeostasis [67, 68].”
Please see Discussion section, lines 481-488 and new references 67 and 68.
- The study mentions that Cnt and C2Cnt increase oxidative stress, leading to CRC cell death. However, was a ROS scavenger, such as N-acetylcysteine (NAC), used to confirm that oxidative stress is a direct cause of cell death?
Reply: According to critical suggestion, new experiments have been performed using the ROS scavenger N-acetylcysteine (NAC), in order to assess if oxidative stress is a direct cause of cell death. “In addition, pre-treatment with ROS scavenger NAC reduced the autophagy activation and apoptosis induced by Cnt+C2Cnt (Figure S3), confirming the role of oxidative stress in mediating cell death mechanism(s) in CRC cells [16]."
Please see Results section, lines 355-358 and new data in Supplementary Figure S3.
Our study highlighted that the induction of apoptosis and autophagy by Cnt and C2Cnt is strongly linked to oxidative stress. The use of NAC, a well-known antioxidant, reduced autophagic flux and cell death, suggesting that ROS generation plays a crucial role in me-diating these effects. These findings align with previous studies indicating that metabolic stress and mitochondrial dysfunction in CRC cells can promote both autophagy and apoptosis through ROS-dependent mechanisms [16].
Please see Results section, lines 384-390.
- Check the data in the control group of Figure 1C and 1H. Each point means data in the control group. The four data of the control group can’t be the same.
Reply: Authors thank the Reviewer for the comment. In figure 1C and 1H the four data of control group were reported as a percentage of the control (Ctr 100%). According to Reviewer comment, the mean of optical density has been also provided in the Revised Manuscript to enhance clarity and accuracy.
Please see new Supplementary Figure S1.
- Compared to Figure 3G with 3L, the background of the confocal images is different. It’s better to keep the same conditions for every confocal image.
Reply: Authors thank the Reviewer for the observation. As requested, microscopy fluorescent images have been adjusted to maintain the same condition across images.
Please, see Figure 3G and 3L.
- It is recommended to enlarge Figure 4J and 4L as it is difficult to read the details.
Reply: Authors thank the Reviewer for the suggestion. Figure 4J and 4L sizes have been enlarge to improve readability of details.
Please, see Figure 4J and 4L.